# Biotransformation of Chlorpyrifos *Shewanella oneidensis* MR-1 in the Presence of Goethite: Experimental Optimization and Degradation Products

**DOI:** 10.3390/toxics12060402

**Published:** 2024-05-31

**Authors:** Shen Tang, Yanhong Li, Zongqiang Zhu, Yaru Wang, Yuqing Peng, Jing Zhang, Peijie Nong, Shufen Pan, Yinming Fan, Yinian Zhu

**Affiliations:** 1College of Environmental Science and Engineering, Guilin University of Technology, No. 319, Yanshan District, Guilin 541004, China; tangshen@glut.edu.cn (S.T.); wanglyywxd@163.com (Y.W.); pengyuqing@glut.edu.cn (Y.P.); zj@glut.edu.cn (J.Z.); nongpeijie09@163.com (P.N.); shufen.pan@mubon.com.cn (S.P.); 2Guangxi Key Laboratory of Environmental Pollution Control Theory and Technology, Guilin University of Technology, Guilin 541004, China; zhuzongqiang@glut.edu.cn; 3Collaborative Innovation Center for Water Pollution Control and Water Safety in Karst Area, Guilin University of Technology, Guilin 541004, China; zhuyinian@glut.edu.cn

**Keywords:** mineral, pesticide, *Shewanella* sp., biodegradation, contamination

## Abstract

In this study, the degradation system of *Shewanella oneidensis* MR-1 and goethite was constructed with chlorpyrifos as the target contaminant. The effects of initial pH, contaminant concentration, and temperature on the removal rate of chlorpyrifos during the degradation process were investigated. The experimental conditions were optimized by response surface methodology with a Box–Behnken design (BBD). The results show that the removal rate of chlorpyrifos is 75.71% at pH = 6.86, an initial concentration of 19.18 mg·L^−1^, and a temperature of 30.71 °C. LC-MS/MS analyses showed that the degradation products were C_4_H_11_O_3_PS, C_7_H_7_Cl_3_NO_4_P, C_9_H_11_Cl_2_NO_3_PS, C_7_H_7_Cl_3_NO_3_PS, C_9_H_11_Cl_3_NO_4_P, C_4_H_11_O_2_PS, and C_5_H_2_Cl_3_NO. Presumably, the degradation pathways involved are: enzymatic degradation, hydrolysis, dealkylation, desulfur hydrolysis, and dechlorination. The findings of this study demonstrate the efficacy of the goethite/*S. oneidensis* MR-1 complex system in the removal of chlorpyrifos from water. Consequently, this research contributes to the establishment of a theoretical framework for the microbial remediation of organophosphorus pesticides in aqueous environments.

## 1. Introduction

Chlorpyrifos, a moderately toxic and moderately persistent organophosphate insecticide, is extensively utilized worldwide. Due to its widespread application in crop pest management, residues of chlorpyrifos have been detected in various regions, including water, soil, and agricultural products [1]. The presence of chlorpyrifos in water bodies has been found to have detrimental effects on the growth and development of aquatic organisms. Additionally, it has been observed that chlorpyrifos can enter the human body through the consumption of contaminated drinking water and food, thereby posing a significant risk to human health [2,3]. In a study examining the behavior and pathology of chlorpyrifos on freshwater ponytail (*Channa punctatus*), it was found that, when freshwater ponytail (*Channa punctatus*) was kept in water containing chlorpyrifos (5 ppm) for 30 days, it showed greater damage to the gills, liver, and intestines, which severely impacted the growth and survival of the fish [4]. Chlorpyrifos has the potential to infiltrate the human body via dietary intake, inhalation, and dermal exposure. Its absorption into the circulatory system can transpire swiftly, leading to the manifestation of adverse health effects within a timeframe ranging from minutes to two hours subsequent to ingestion. The protracted exposure to chlorpyrifos has been associated with a range of adverse effects, such as liver dysfunction, hematotoxicity, immunological abnormalities, embryotoxicity, genotoxicity, teratogenicity, as well as neurochemical and behavioral alterations [5]. Therefore, it is necessary to understand the transformation and degradation of chlorpyrifos in different environments.

In the natural environment, microorganisms can use organic matter, even toxic organic matter, as a carbon source to obtain energy and synthesize other compounds they need. Over the past decade, researchers have used this property of microbes to study the degradation of organic pollutants in the environment [6]. This technology, which uses the inherent degradation ability of microorganisms to decompose complex organic pollutants into simpler, low-molecular-weight organic compounds, thereby promoting water purification, is called microbial remediation technology. Various microorganisms, such as *Bacillus*, *Pseudomonas*, *Klebsiella*, *Shewanella*, and *Geobacter* sp., have been utilized in studies investigating the degradation of pesticide residues [7,8]. *Shewanella* sp., a facultative anaerobic bacterium, is commonly distributed in both soil and aquatic sediments. *Shewanella* sp. can not only degrade a variety of organic pollutants, but also reduce heavy metal ions through extracellular electron transfer [9]. Therefore, *Shewanella* sp. has become the most popular pollutant degrading strain in current research. Govarthanan et al. used *Shewanella* BT05 to degrade chlorpyrifos at different concentrations of 10 to 50 mg·L^−1^. The degradation rates observed after 24 h were 94.3%, 91.8%, 87.9%, 82.6%, and 80.5% for the respective concentrations. To optimize the experimental conditions, a response surface methodology-centered composite design method was utilized, resulting in a 93% removal of chlorpyrifos. This achievement was attained under specific conditions, namely a pH of 7.0, a temperature of 32.5 °C, and a glucose concentration of 3.5% [10].

For highly toxic pesticide pollutants, the efficiency of microbial degradation alone is low. In order to further improve the degradation ability of *Shewanella* sp., various nanomaterials and minerals have been applied to the degradation system [8,11,12]. Studies have shown that nanomaterials are not friendly to aquatic organisms, which limits the application of nanomaterials [13,14]. The interaction of *Shewanella* sp. with minerals is complex. In contact with different minerals, *Shewanella* sp. produces different stress responses, resulting in the secretion of extracellular polymers of different compositions. The addition of minerals with redox activity, such as iron and manganese compounds, to a bioelectrochemical degradation system can change the composition of the extracellular polymers. Changes in the composition of extracellular polymers can lead to changes in the extracellular electron transfer efficiency of *Shewanella* sp., thereby altering the degradation efficiency of pollutants [9,15]. Zheng et al. found that *S. oneidensis* MR-1 synergistically with goethite and riboflavin (RF) could enhance Cr (VI) bioreduction [16]. In the Tang et al. study, anthraquinone-2-sulfonate (AQDS) was found to significantly promote Fe (III) bioreduction by *S. oneidensis* MR-1, and different concentrations of aniline were added to the system for degradation. The results showed that the best degradation of aniline was achieved when the aniline concentration was 3 μM [17]. Therefore, whether the degradation system constructed by goethite and *S. oneidensis* MR-1 can be applied to the degradation of chlorpyrifos in water is a problem that has not been fully studied.

However, chlorpyrifos degradation studies should consider not only its initial concentration and degradation efficiency, but also its degradation products. Microbial degradation is considered to be the conversion of target pollutants by microorganisms into carbon and energy sources required for their growth and reproduction, which are involved in microbial metabolism [18,19,20]. However, higher concentrations of organic pollutants, especially toxic pollutants, tend to inhibit microbial reproduction and metabolism, resulting in low microbial degradation efficiency. Gu’s group proposed a new concept of microbial degradation: toxicity, metabolism, and maintenance energy [21]. These researchers argue that: when the concentration of toxic pollutants is low, microorganisms can convert toxic pollutants into carbon and energy for maintaining their own growth and metabolism; when the concentration of toxic pollutants is high, the pollutants show toxic side effects on microorganisms, and the growth and metabolism of microorganisms are inhibited. Therefore, only when the concentration of toxic pollutants is within a reasonable range can microorganisms efficiently transform these pollutants. Chlorpyrifos and its metabolite TCP not only have insecticidal properties, but also hinder the proliferation of chlorpyrifos-degrading bacteria [22]. In addition, the coexistence of TCP with parental organisms leads to toxic synergies [23]. Although TCP, the main metabolite of chlorpyrifos, has been found to be less toxic than chlorpyrifos itself, its degradation process is much longer [24]. Despite the progress in the study of microbial degradation of organic pollutants, the degradation effect and degradation products will be affected by microbial diversity, exogenous material interference, and changes in the external environment. Therefore, there is a need for an in-depth study of the optimal experimental conditions for microbial degradation systems, including factors such as temperature and initial concentration of pollutants, as well as the effects of coexisting substances [1,18]. Such a comprehensive exploration is a prerequisite and foundation for a deeper understanding of the degradation mechanisms and warrants further research.

This study presents the construction of a goethite/*Shewanella oneidensis* MR-1 degradation system, employing *S. oneidensis* MR-1 as the bacterial strain responsible for degradation. The optimization of experimental parameters, including pollutant concentration, degradation temperature, and initial pH, was conducted using both response surface methodology (RSM) and LC-MS/MS analysis to identify the degradation products. The identification of degradation products was accomplished using LC-MS/MS in this study. This research contributes to the understanding of microbial degradation conditions and degradation products associated with organophosphorus pesticides, thereby offering valuable insights for the investigation of water pollution caused by organophosphorus pesticides.

## 2. Materials and Method

### 2.1. Chemicals and Reagents

Analytical reagent-grade chemicals, including anthraquinone-2,6-disulfonic acid (AQDS), Fe (NO_3_)_3_·9H_2_O, NaOH, HCl, and NaCl, were obtained from Shanghai Macklin Biochemical Co., Ltd. (Shanghai, China). Tryptone, yeast extract, agar, sodium L-lactate, and 4-(2-hydroxyethyl)-1-piperazineethanesulfonic acid (HEPES) were obtained from Bioengineering (Shanghai) Co., Ltd. (Shanghai, China). The analytical standard for chlorpyrifos (≥99.9%) was acquired from Shanghai Anpel Experimental Technology Co., Ltd. (Shanghai, China).

### 2.2. Synthesis of Goethite

Goethite was synthesized following a modified procedure based on previous research [25]. Firstly, 20 g of Fe (NO_3_)_3_·9H_2_O was added to a 500 mL sterilized beaker and then dissolved with 200 mL of deionized water. The resulting solution was placed on a magnetic stirrer with continuous stirring for 24 h. Then, 1.0 mol·L^−1^ of NaOH was added dropwise until the pH reached 12 and was placed in a 60 °C thermostat for 5 days. The obtained thick suspension was dried in an oven at 60 °C.

### 2.3. S. oneidensis MR-1 and Its Culture Condition

#### 2.3.1. Strain and Culture Methods

The strain *S. oneidensis* MR-1 was purchased from the Marine Culture Collection of China (MCCC, ATCC 700550).

The methods for culturing the strain and measuring its optical density values were carried out as described in a previous study [26]. Strains were grown in Luria–Bertani (LB) solid medium (casein tryptone 10.0 g·L^−1^, yeast extract fermentation 5.0 g·L^−1^, NaCl 10.0 g·L^−1^, agar 15 g·L^−1^, and pH adjusted to 7). To obtain the bacterial cultures required for the experiments, cells were inoculated into LB liquid medium containing casein trypsin (10.0 g·L^−1^), yeast extract fermentation (5.0 g·L^−1^), and NaCl (10.0 g·L^−1^). The pH was adjusted to 7. The cultures were then shaken continuously for 24 h at 160 rpm in an oscillating incubator at 35 °C until they reached the logarithmic growth phase.

After shaking for 24 h, cells were collected by centrifugation (4000 rpm, 4 °C, and 10 min), washed three times with sterile 30 mM HEPES buffer, and then resuspended in the same concentration of HEPES buffer for use in subsequent experiments. The optical density of the bacterial solution was measured using a UV-Vis spectrophotometer (Shanghai Metash Instruments Co., Ltd., UV-9000S, Shanghai, China) with absorbance at 600 nm (OD_600_).

#### 2.3.2. Acclimated Processes of *S. oneidensis* MR-1

Acclimated processes of *S.oneidensis* MR-1 were modified according to the previous report [27]. To acclimate the strain culture, 10 mL of the culture (OD_600_ = 1.0) was first transferred to chlorpyrifos medium at 5 mg·L^−1^ and incubated in an oscillating incubator at 35 °C for 24 h. Subsequently, the culture was transferred to chlorpyrifos medium at a concentration of 10 mg·L^−1^ and incubated in an oscillating incubator at 35 °C for 24 h. The remaining acclimated processes followed the steps described above, with concentration gradients of 20, 30, 40, and 50 mg·L^−1^.

### 2.4. Design and Optimization of Chlorpyrifos Degradation Experiment

The experimental scheme is shown in Figure 1. The pH was adjusted to 5, 6, 7, 8, and 9 with 0.5 mol·L^−1^ of NaOH or HCl. Nitrogen was passed into the flask to create an anaerobic environment and shaken in a constant-temperature shaker at 30 °C (160 rpm). Samples were taken periodically for analysis. All experiments were carried out in triplicate.

Detailed experimental process: 2 mg of chlorpyrifos was added to the conical bottle, dissolved in petroleum ether, and the solvent was blown dry with nitrogen. Add 50 mL of ultrapure water and oscillate on a cyclotron oscillator for 1 h (160 rpm) to obtain chlorpyrifos suspension. Add 5 mL of LB medium, 30 mmol of sodium lactate, and 10 mg of AQDS successively. After inoculating 20 mL of *S. oneidensis* MR-1 bacterial suspension (OD_600_ = 1.0), 40 mg of goethite was added and 50 mL of ultrapure water was added. The system was oscillated on a cyclotron oscillator and sampled periodically for determination.

The response surface methodology (RSM) in Design-Expert (version 12) software was used to optimize the experimental conditions for chlorpyrifos degradation by *S. oneidensis* MR-1, including the key factors and the interactions between them. The independent variables selected for this study were initial pH, temperature, and degradation time (Table 1). A Box–Behnken design (BBD) was employed to generate 17 sets of experiments. The data were analyzed using the response surface regression procedure, which enabled the fitting of the following quadratic polynomial equation (Equation (1)):(1)$$Y=b_{0}+\alpha_{1}X_{1}+\alpha_{2}X_{2}+\alpha_{3}X_{3}+\beta_{11}X_{1}^{2}+\beta_{22}X_{2}^{2}+\beta_{33}X_{3}^{2}+\beta_{12}X_{1}X_{2}+\beta_{13}X_{1}X_{3}+\beta_{23}X_{2}X_{3}$$
where *Y* is the predicted response value (chlorpyrifos removal rate); *b*_0_ is constant; *X*_1,_
*X*_2_, and *X*_3_ are independent factors; α_1_, α_2_, and α_3_ are linear coefficients; β_11_, β_22_, and β_33_ are quadratic coefficients; and β_12_, β_13_, and β_23_ are cross-product coefficients. Design-Expert (version 12) was used to conduct correlation coefficient (R^2^), analysis of variance (F-value and *p*-value), and draw graphs for the results, and quantify the goodness of fit between the model and the experimental data to verify the model.

### 2.5. Analytical Method

#### 2.5.1. Sample Characterization

Scanning electron microscopy (SEM) was used to observe the cell morphology. Bacteria and bacterial-goethite mixture were collected and rinsed three times with a phosphate buffer solution and then fixed with a 2.5% (*v*/*v*) glutaraldehyde solution for 12 h. The samples were then dehydrated with ethanol at concentrations of 30%, 50%, 70%, 90%, and 100% (*v*/*v*) for 15 min, followed by freeze-drying for 8 h. The bacteria were finally observed with a scanning electron microscope (JSM-7900F, JEOL™, Tokyo, Japan).

Fourier transform infrared spectroscopy (FT-IR) was utilized to detect functional groups on the surface of goethite/bacterial cells. The bacterial-goethite mixture samples were lyophilized at different stages of chlorpyrifos degradation. KBr was mixed with the samples in the ratio of 100:1 and finely ground in an agate mortar. The samples were compressed into flakes under 15 MPa pressure and then placed in the sample chamber of an infrared spectrometer (PerkinElmer Frontier, Waltham, MA, USA) to obtain infrared spectra. The scanning wave number range was in the range of 400–4000 cm^−1^.

To characterize the mineral surfaces of the synthesized goethite minerals, X-ray power diffraction (XRD) analysis was carried out (PANalytical X’Pert, Malvern Panalytical Ltd., Cambridge, UK). X-ray diffraction patterns were obtained in the angular range of 15–65° in steps of 0.02°·s^−1^. X-ray photoelectron spectroscopy (Thermo ESCALAB 250Xi, Waltham, MA, USA) analysis was conducted to determine goethite oxidation states. The data obtained from X-ray powder diffraction and X-ray photoelectron spectroscopy analyses were analyzed using Avantage (Thermo Fisher Scientific, version 5.9922).

#### 2.5.2. Analysis of Chlorpyrifos

To monitor the concentration of chlorpyrifos in the samples during the experiment, samples were taken regularly and pretreated by liquid–liquid extraction. High-performance liquid chromatography (HPLC) was used to analyze chlorpyrifos concentration using a ZORBAX Rx-C18 column (4.6 × 150 mm, 5 µm) at a column temperature of 30 °C. The mobile phase consisted of ultrapure water (A) and methanol (B) in a ratio of A:B = 20:80, with a flow rate of 1 mL·min^−1^ and an injection volume of 10 μL. The UV detector wavelength was set to 230 nm. Three parallel experiments were conducted in each group, and the experimental results were averaged.

The degradation products of chlorpyrifos were analyzed by a high-resolution mass spectrometer (LC-Q-TOF-MS/MS), with an injection volume of 5 μL, a flow rate of 0.3 mL·min^−1^, and Waters BEH C18(2.1 × 50 mm, 1.7 μm) as the chromatographic column. The liquid chromatography elution procedure is described in Appendix A. The ion source gas temperature was set to 350 °C and the ion gas concentration was 12 mL·min^−1^. Electrospray ion source positive-mode (ESI+) voltage was set to 4000 V; electrospray ion source negative-mode (ESI-) voltage was set to 3200 V. The mass spectrum collision energy was set to 175 V.

## 3. Results and Discussion

### 3.1. Experimental Optimization Results and Model Analysis

RSM is an efficient experimental optimization method that can quickly determine the optimal response value (i.e., removal rate) by adjusting and optimizing each parameter. The Box–Behnken design is one of the most important RSM designs used to determine optimal conditions [28,29]. This design considers the interactions between parameters and requires experiments at only three levels, reducing the number of time-consuming and labor-intensive experiments needed to gather sufficient data [30]. To assess trial layouts and build a mathematical model, the statistical program Design Expert (Stat-Ease, Minneapolis, MN, USA) was used.

In this study, 17 experimental sets were established using the Box–Behnken design to determine the effects of initial pH, temperature, and degradation time on chlorpyrifos removal rate. The results of the experiments are presented in Table 2.

To determine the significance of the second-order polynomial equation fit, an analysis of variance (ANOVA) was conducted on the experimental data. The results of this analysis are presented in Table 3.

The results of the ANOVA analysis reveal that the experimental design is reliable and statistically significant (F = 52.69, *p* < 0.0001). The analysis also shows that the model has good confidence (lock of fit *p* = 0.1145). Moreover, the model’s correlation coefficient (R^2^ = 0.9855) and the corrected coefficient of determination (Adjusted R^2^ = 0.9667) indicate a good fit between the expected and actual values (Figure 2a).

Furthermore, $X_{3}$, $X_{2}^{2},$ and $X_{3}^{2}$ showed no significant difference (*p* > 0.05), $X_{1}$ showed a highly significant difference (*p* = 0.0034), and $X_{2}$ and $X_{1}^{2}$ showed no significant difference (*p* < 0.05). Based on these results, it can be concluded that $X_{1}$ > $X_{2}$ > $X_{3}$ are the factors influencing the removal rate of chlorpyrifos in the degradation experiment. Figure 2c,d shows that there is no interaction (*p* < 0.05) between the pH and initial concentration of chlorpyrifos (${X_{1}X}_{2}$) and between the initial concentration of chlorpyrifos and temperature (${X_{2}X}_{3}$). However, there was an interaction (*p* < 0.05) between pH and temperature (${X_{1}X}_{3})$, as shown in Figure 2b.

Multiple regression analysis was conducted on the experimental results, and the second-order polynomial equation of chlorpyrifos removal efficiency (Equation (2)) was obtained as follows:(2)$$Y=72.50-4.38\;X_{1}-3.48\;X_{2}+1.22\;X_{3}-0.6975\;X_{1}X_{2}+3.82\;X_{1}X_{3}+0.5715\;X_{2}X_{3}-26.17\;X_{1}^{2}-10.21\;X_{2}^{2}-3.04\;X_{3}^{2}$$
where *Y* is the predicted value of the chlorpyrifos removal rate; $X_{1}$, $X_{2},$ and $X_{3}$ are the coding terms of three independent test variables, initial pH, initial concentration of chlorpyrifos, and temperature, respectively. The optimal values of the selected experimental variables were obtained using Equation (1) and analyze the response surface contour plots.

The optimal experimental conditions for chlorpyrifos degradation were determined to be pH = 6.86, initial concentration C = 19.18 mg·L^−1^, and degradation temperature T = 30.71 °C, based on Equation (2). Using these conditions in the response surface method, the degradation rate of chlorpyrifos is 75.71%, which is very close to the predicted value of 74.41%.

### 3.2. Characterization of Goethite

#### 3.2.1. SEM Analysis

The synthesized goethite nanoparticles are needle-like in shape and exhibit good crystallinity. The nanoparticles exhibited a uniform size (≈600 nm in length and 70 nm in width) and displayed some degree of agglomeration (Figure 3a). The SEM images reveal that the cells of *S. oneidensis* MR-1 are elongated rods, measuring approximately 2–3 μm in length and 0.32 μm in diameter (Figure 3b). Upon domestication with chlorpyrifos, the physiological traits of *S. oneidensis* MR-1 are affected, and its individuals become stout (Figure 3c), measuring approximately 2–3 μm in length and 0.75 μm in diameter. Upon domestication with varying concentrations of chlorpyrifos, the external morphology of *S. oneidensis* MR-1 was altered while still retaining its rod-shaped form. Notably, the diameter of the cells increased up to two-fold when compared to the wild-type *S. oneidensis* MR-1 (Figure 3b,c). These morphological changes may be attributed to *S. oneidensis* MR-1 as it adapts to the toxicity of chlorpyrifos. *S. oneidensis* MR-1 has similar morphological changes in response to the toxicity of heavy metals [31]. Figure 3d shows an electron microscope image of goethite/*S. oneidensis* MR-1 after degradation. The image reveals that goethite was attached to domesticated *S. oneidensis* MR-1 in the goethite/*S. oneidensis* MR-1 complex after chlorpyrifos degradation.

#### 3.2.2. FT-IR Analysis

The degradation of chlorpyrifos by *S. oneidensis* MR-1 was analyzed using FT-IR spectroscopy, and the results are presented in Figure 4 and Appendix A. Before degradation, goethite had broad peaks of high intensity at 3100–3500 cm^−1^. Among these peaks, the 3426 cm^−1^ peak was attributed to the stretching vibration formed by the hydroxyl group (−OH) on the surface of goethite combined with water molecules (H-OH) [32,33], while the 1645 cm^−1^ peak was due to the C=O stretching vibration [34]. Additionally, due to the adsorption of phosphate and organic phosphorus pollutants [35,36], a broad peak in the 1360–1450 cm^−1^ range was observed at the start of the degradation. This may be related to the tensile vibration of the pyridine ring in chlorpyrifos molecules adsorbed on the goethite surface.

Typically, the pyridine ring in the chlorpyrifos molecule has a distinct peak at 1412 cm^−1^ [37], which gradually decreases in intensity as the degradation progresses [10]. The observation of a weak stretching vibrational peak at 1051 cm^−1^ is generally considered to be the vibrational peak formed by FeO-P-OFe [38]. When the purity of goethite is high, the O-H bending vibration peak of the Fe-OH group appears at 890 cm^−1^ and 796 cm^−1^, and the tensile vibration peak of Fe-O appears at 637 cm^−1^ [38,39].

#### 3.2.3. XRD Analysis

XRD analysis was conducted on goethite before and after chlorpyrifos degradation, and the results (Figure 5) were compared with the goethite standard card (JCPDS 00-29-0713). The analysis revealed that the experimentally synthesized goethite crystals were partially agglomerated, but had smooth surfaces and high crystallinity. The synthesized goethite contained iron hydroxide and the diffraction peaks of iron hydroxide disappeared after degradation. The reduction of Fe (III) in the Fe (III) material by *S. oneidensis* MR-1 starts from iron hydroxide, which is consistent with the experimental results of Guo’s group [40].

#### 3.2.4. XPS Analysis

XPS spectroscopic studies were carried out on the goethite before and after degradation, and the results are shown in Figure 6. The full spectral scans of the XPS spectra of the goethite at the end of degradation (day 10) and at the beginning of degradation (day 2) are shown in the top and bottom parts of Figure 6a, respectively. The top and bottom parts of Figure 6b–d show: the deconvoluted peaks of C1s demonstrate (Figure 6b) that, by day 2 of degradation, there are higher levels of hydrocarbons (C-C/C=C, 25.54%), hydroxyl (C-OH, 17.13%), carbonyl (C=O, 26.06%), and carboxyl (O-C=O, 6.26%) groups, which could be organic compounds, such as chlorpyrifos secreted by *S. oneidensis* MR-1 extracellular polymer and adsorbed by goethite. This result is consistent with the FT-IR analysis. When degradation proceeded to day 10, chlorpyrifos molecules were broken down into smaller molecules, and the hydrocarbon (C-C/C=C, 15.30%), hydroxyl (C-OH, 8.89%), and carbonyl (C=O, 6.57%) groups on the surface of the goethite were reduced. The carboxyl group (O-C=O) was transferred to the aqueous phase, as it is highly soluble in water. During this process, the binding energies of the hydrocarbons (C-C/C=C, 284.64 eV), hydroxyl groups (C-OH, 285.63 eV), and carbonyl groups (C=O, 287.02 eV) shifted to lower values (Appendix A). Some studies suggest that this shift may make the Fe center more Lewis acidic and may accelerate the Fe (III)/Fe (II) redox cycle, increasing electron transfer [41,42,43,44].

To observe changes in the hydroxyl groups on the goethite surface, changes in the binding energy of O1s were monitored using XPS spectroscopy (Figure 6c and Appendix A). The peaks at 529.86 eV, 531.73 eV, and 533.48 eV correspond to the binding energies of Fe-O, Fe-OH, and adsorbed H_2_O, and are typical of goethite, in agreement with previously reported results [45].

XPS spectra of Fe (III) species could be decomposed into two intervals, Fe 2p1/2 and Fe 2p3/2, with binding energies of 724.18 eV, 710.88 eV and 725.62 eV, 712.78 eV, respectively; the satellite peak was at 719.88 eV (Appendix A). At the beginning of degradation (day 2), Fe (II) species were not detected on the goethite surface. At the end of degradation (day 10), 3.94% of Fe (II) species was detectable on the goethite surface (Appendix A). The peaks at 709.54 eV and 722.75 eV were attributed to Fe (II) species [45]. This result indicates the coexistence and transformation of Fe (III) and Fe (II) species during the degradation process. It can be inferred that electrons generated during the metabolism of *S.oneidensis* MR-1 are transferred to chlorpyrifos molecules through the reversible redox of Fe (III) species and Fe (II) on the surface of goethite, forming a Fenton-like reaction that facilitates the decomposition of chlorpyrifos molecules [43,46].

### 3.3. Interference of Degradation Products and Pathways

The degradation products of chlorpyrifos were identified by high-resolution mass spectrometry (LC-Q-TOF-MS/MS). The analytical results show that the primary mass spectra (MS) of chlorpyrifos (C_9_H_11_Cl_3_NO_3_PS) metabolites have very distinct excimer ion peaks (*m*/*z* = 349. 97). The characteristic peaks of fragment ions obtained by secondary mass spectrometry (MS/MS) were *m*/*z* = 169.08, 308.29, 315.28, 320.95, 335.63, 152.00, and 197.93, with molecular formulae C_4_H_11_O_3_PS, C_7_H_7_Cl_3_NO_4_P, C_9_H_11_Cl_2_NO_3_PS, C_7_H_7_Cl_3_NO_3_PS, C_9_H_11_Cl_3_NO_4_P, C_4_H_11_O_2_PS, and C_5_H_2_Cl_3_NO, respectively. The metabolite names are provided in Table 4.

The microbial degradation reaction of chlorpyrifos is typically divided into enzymatic and non-enzymatic degradations [7,22,47]. Enzymatic degradation involves the breakdown of P-O or P=S bonds in chlorpyrifos by organophosphorus-degrading enzymes, which reduces its toxicity [1]. Based on the results of MS/MS mass spectrometry, it was inferred that, during the degradation of chlorpyrifos by goethite/*S. oneidensis* MR-1, Compound 6 (Chlorpyrifos, *m*/*z* = 349.97) broke the P-O bond catalyzed by organophosphorus degrading enzymes to form the less toxic metabolites Compound 8 (3,5,6-Trichloro-2-pyridinol, *m*/*z* = 197.93) and Compound 7 (O,O-diethyl thiophosphonate, *m*/*z* = 152.00). Typically, 3,5,6-trichloro-2-pyridinol (TCP) is the major metabolite of Chlorpyrifos in soil, microorganisms, plants, and humans [48]. Compound 7 (O,O-diethyl phosphorothioate, *m*/*z* = 152.00) was further hydrolyzed by hydrolase to form Compound 1 (O,O-diethyl phosphorothioate, *m*/*z* = 169.08) [1]. Compound 6 (Chlorpyrifos, *m*/*z* = 349.97) can also be dealkylated to Compound 4 (Chlorpyrifos-methyl, *m*/*z* = 320.95) by removing two methylenes each. During catabolic degradation, some microorganisms can utilize pesticides as a source of carbon and energy. The presence of Compound 4 was detected in this study, indicating that some chlorpyrifos molecules can be converted by *S. oneidensis* MR-1 into their own utilizable carbon and energy sources during the degradation of chlorpyrifos by goethite/*S. oneidensis* MR-1 [48]. Compound 6 (Chlorpyrifos, *m*/*z* = 349.97) breaks the P = S double bond in the molecule, losing a sulfur atom and gaining an oxygen atom by hydrolysis to form Compound 5 (Chlorpyrifos oxon, *m*/*z* = 335.63). This is an unstable intermediate formed by chlorpyrifos through oxidative desulfurization or acylation [49,50]. Compound 2 (Chlorpyrifos methyl oxon, *m*/*z* = 308.29) is formed when the P = S double bond in the molecule of Compound 4 (Chlorpyrifos-methyl, *m*/*z* = 320.95) breaks, losing a sulfur atom and hydrolyzing to produce an oxygen atom. Compound 3 (Dechlorination of Chlorpyrifos, *m*/*z* = 315.28) is formed when Compound 6 (Chlorpyrifos, *m*/*z* = 349.97) loses a chlorine atom from the pyridine ring in the presence of dichlorination [51,52].

Therefore, at least five degradation pathways exist in the degradation of chlorpyrifos by goethite/*S. oneidensis* MR-1: (1) enzymatic degradation; (2) hydrolysis; (3) dealkylation; (4) desulfur hydrolysis; and (5) dechlorination, as shown in Figure 7.

Comparing the microbial degradation studies of chlorpyrifos in recent years, it is easy to find that goethite/*S. oneidensis* MR-1 does not have an advantage in terms of degradation efficiency and chlorpyrifos tolerance (Table 5). However, the goethite/*S. oneidensis* MR-1 degradation system produced more types of degradation products during the degradation process. This may be related to the abundant extracellular electron transfer channels and metabolic pathways of *S. oneidensis* MR-1. More and deeper research are needed on and into these issues.

## 4. Conclusions

In this paper, the degradation conditions and degradation products of chlorpyrifos by the goethite/*S. oneidensis* MR-1 degradation system were investigated using the RSM-BBD method to optimize the three independent parameters of initial pH, initial concentration of chlorpyrifos, and temperature. The optimal conditions for the experiments were found to be pH = 6.86, chlorpyrifos concentration = 19.18 mg·L^-1^, and temperature = 30.71 °C. The predicted removal rate of chlorpyrifos was 74.41% and the actual removal rate was 75.71%; the predicted value coincided with the measured value. FT-IR and XRD studies confirmed that *S. oneidensis* MR-1 reduced Fe (III) to Fe (II) in the goethite mineral. Goethite not only adsorbed chlorpyrifos, but also acted as a mediator of extracellular electron transfer, which enhanced the degradation of chlorpyrifos by *S. oneidensis* MR-1. Furthermore, XPS analysis confirmed that *S. oneidensis* MR-1 acts as a reducing agent for Fe (III) on the goethite surface. This electron transfer mediated by goethite plays an important role in promoting the degradation of chlorpyrifos. LC-Q-TOF-MS/MS analysis identified seven degradation products: C_4_H_11_O_3_PS, C_7_H_7_Cl_3_NO_4_P, C_9_H_11_Cl_2_NO_3_PS, C_7_H_7_Cl_3_NO_3_PS, C_9_H_11_Cl_3_NO_4_P, C_4_H_11_O_2_PS, and C_5_H_2_Cl_3_NO. It was inferred that the degradation of chlorpyrifos molecules undergoes at least five pathways: enzymatic degradation, hydrolysis, dealkylation, desulfur hydrolysis, and dechlorination. Although the goethite/*S. oneidensis* degradation system did not have an advantage in terms of degradation efficiency and degradation rate, the goethite/*S. oneidensis* degradation system produced a wider variety of degradation products. This suggests that Chlamydomonas reinhardtii can utilize its numerous metabolic pathways for the biodegradation of chlorpyrifos. This also suggests that the goethite/*S. oneidensis* degradation system is a candidate for organophosphorus pesticide degradation.

## Figures and Tables

**Figure 1 toxics-12-00402-f001:**
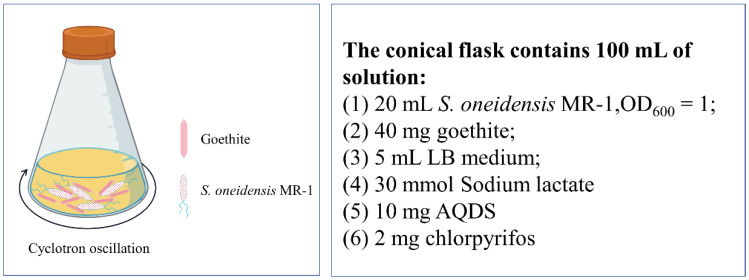
Goethite/*Shewanella oneidensis* MR-1 degradation system.

**Figure 2 toxics-12-00402-f002:**
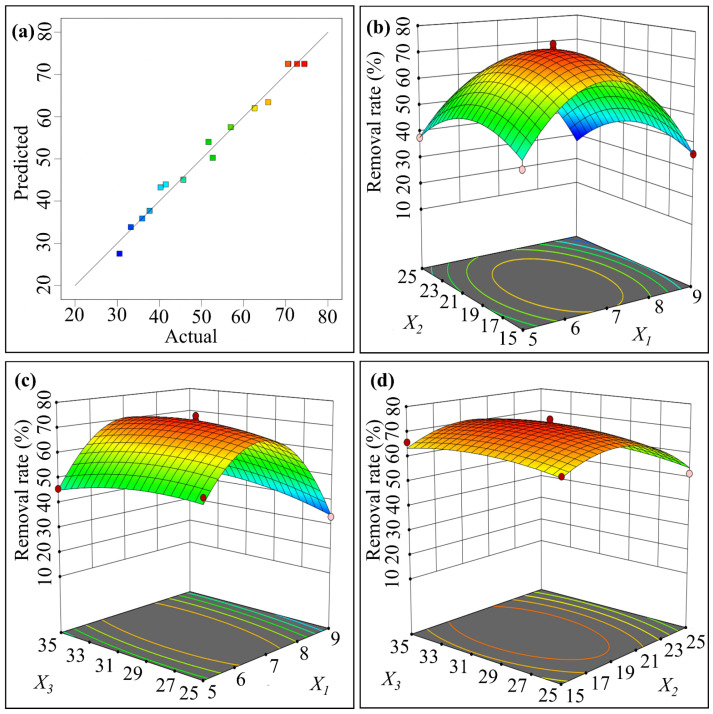
Fit of model predicted values to actual values (**a**) and response surface diagram of the interaction between different factors (**b**–**d**).

**Figure 3 toxics-12-00402-f003:**
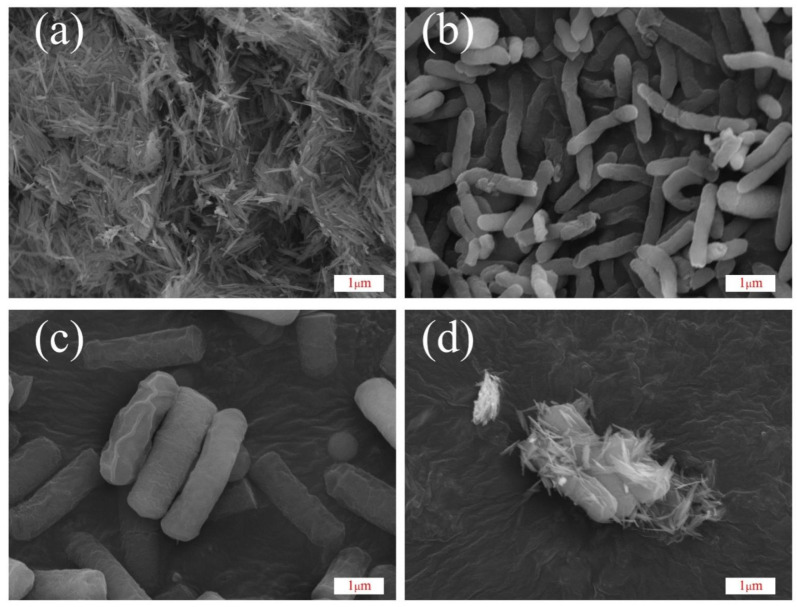
SEM images of (**a**) synthesized goethite, *S. oneidensis* MR-1 (**b**) before and (**c**) after domestication, and (**d**) complex of *S. oneidensis* MR-1 and goethite after degradation of chlorpyrifos.

**Figure 4 toxics-12-00402-f004:**
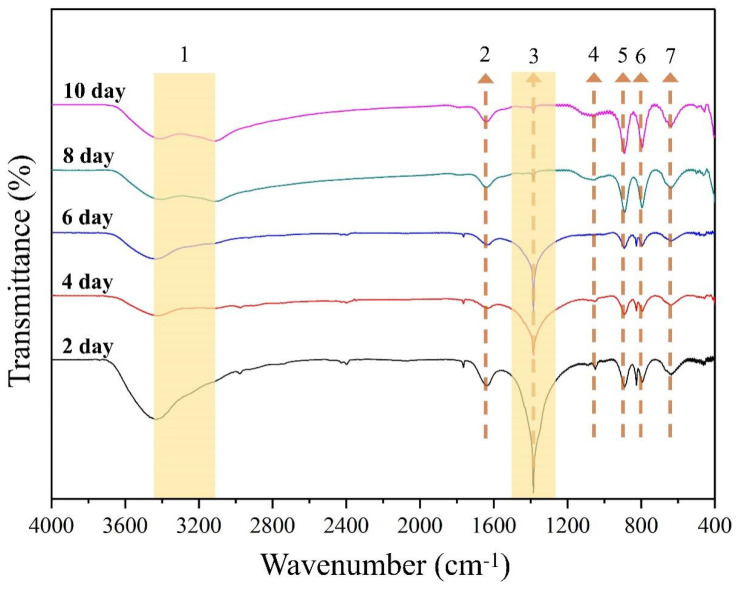
FT-IR spectra of goethite/*S. oneidensis* MR-1 during the degradation of chlorpyrifos. The numbers in the figure indicate peak positions or wavenumber. These wavenumbers represent certain functional groups. The functional groups represented by numbers 1 to 7 have been shown in detail in Appendix A Appendix A.

**Figure 5 toxics-12-00402-f005:**
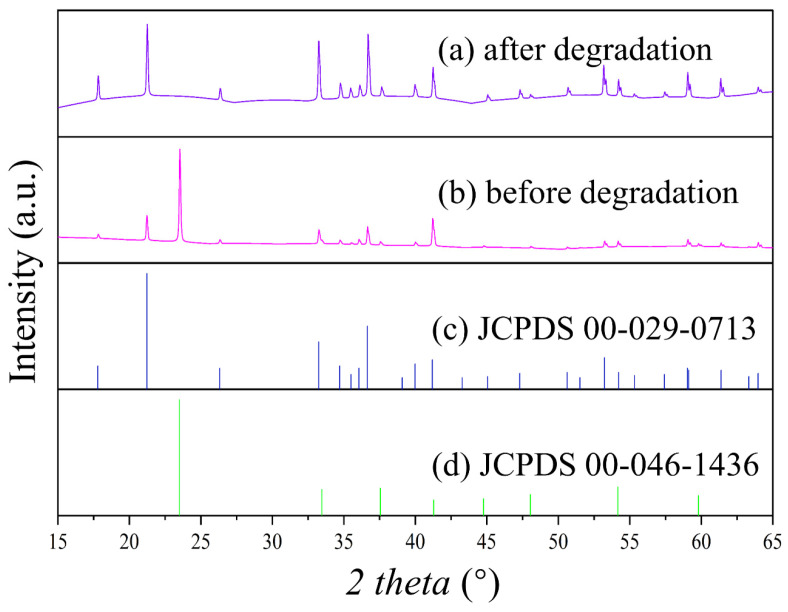
XRD spectra of acicular ferrite (**a**,**b**) before and after chlorpyrifos degradation; XRD spectra of standard goethite (**c**) and standard iron hydroxide (**d**).

**Figure 6 toxics-12-00402-f006:**
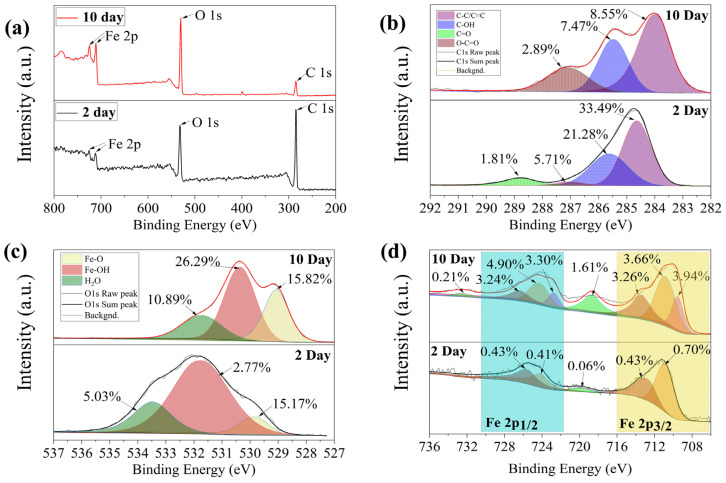
XPS spectra of goethite/*S. oneidensis* MR-1 before and after the degradation of chlorpyrifos. The full spectral scans of the XPS spectra of the goethite at the end of degradation (day 10) and at the beginning of degradation (day 2) are shown in the top and bottom parts of (**a**), respectively. The top and bottom parts of (**b**–**d**) show: the deconvoluted peaks of C1s (**b**) and O1s (**c**).

**Figure 7 toxics-12-00402-f007:**
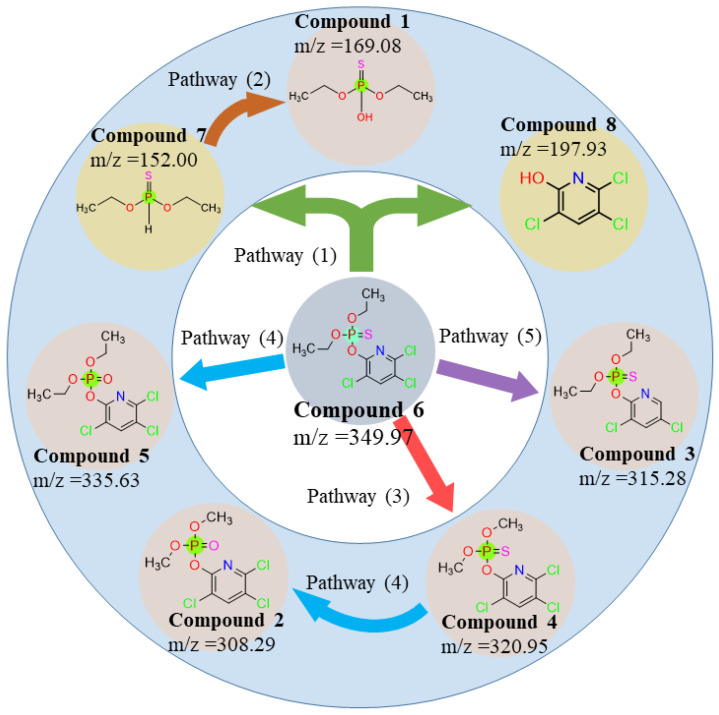
Chlorpyrifos degradation products.

**Table 1 toxics-12-00402-t001:** Experimental levels of independent parameters.

Variable	Units	Experimental Levels		
		Lowest (−1)	Middle (0)	Highest (+1)
$X_{1}$ = Initial pH	/	5	7	9
$X_{2}$ = Concentration	mg·L^−1^	15	20	25
$X_{3}$ = Temperature	°C	25	30	35

**Table 2 toxics-12-00402-t002:** Box–Behnken design matrix chlorpyrifos removal rate.

Run	pH	Concentration (mg·L^−1^)	Temperature (°C)	Chlorpyrifos Removal Rate (%)
1	9	20	35	41.52
2	9	15	30	35.95
3	7	25	35	56.94
4	7	20	30	74.36
5	7	15	25	62.59
6	7	20	30	74.41
7	5	20	25	52.68
8	7	20	30	70.48
9	7	15	35	65.81
10	7	20	30	70.59
11	7	20	30	72.65
12	7	25	25	51.65
13	9	20	25	33.24
14	5	15	30	40.28
15	5	20	35	45.69
16	5	25	30	37.68
17	9	25	30	30.56

**Table 3 toxics-12-00402-t003:** Analysis of variance (ANOVA) of the response surface model for the prediction of chlorpyrifos removal rate.

Source	Sum of Squares	df	Mean Square	F-Value	*p*-Value
Model	3883.14	9	431.46	52.69	<0.0001
$X_{1}$ = pH	153.65	1	153.65	18.76	0.0034
$X_{2}$ = Concentration	96.61	1	96.61	11.8	0.0109
$X_{3}$ = Temperature	12	1	12	1.47	0.2653
${X_{1}X}_{2}$	1.95	1	1.95	0.2376	0.6408
${X_{1}X}_{3}$	58.29	1	58.29	7.12	0.0321
${X_{2}X}_{3}$	1.07	1	1.07	0.1308	0.7283
$X_{1}^{2}$	2884.26	1	2884.26	352.22	<0.0001
$X_{2}^{2}$	438.73	1	438.73	53.58	0.0002
$X_{3}^{2}$	38.98	1	38.98	4.76	0.0655
Residual	57.32	7	8.19		
Lock of Fit	42.46	3	14.15	3.81	0.1145
Pure error	14.86	4	3.71		
C.V. %	5.30				
R^2^	0.9855				
Adjusted R^2^	0.9667				

**Table 4 toxics-12-00402-t004:** MRM transitions [*m*/*z*] for CPF and its degradation of products.

No.	*m*/*z*	Molecular Formula	Ionization Mode	Molecular Weight (g/mol)	Name of Degradation Products
1	169.08	C_4_H_11_O_3_PS	[M − H]^+^	170.17	O,O-Diethyl phosphorothionate
2	308.29	C_7_H_7_Cl_3_NO_4_P	[M+2H]^+^	306.47	Chlorpyrifos methyl oxon
3	315.28	C_9_H_11_Cl_2_NO_3_PS	[M − Cl]^+^	315.28	Dechlorination of Chlorpyrifos
4	320.95	C_7_H_7_Cl_3_NO_3_PS	[M − H]^+^	322.53	Chlorpyrifos-methyl
5	335.63	C_9_H_11_Cl_3_NO_4_P	[M+H]^+^	334.52	Chlorpyrifos oxon
6	349.97	C_9_H_11_Cl_3_NO_3_PS	[M]^+^	350.58	Chlorpyrifos
7	152.00	C_4_H_11_O_2_PS	[M − H]^−^	154.17	O,O-diethyl thiophosphonate
8	196.92	C_5_H_2_Cl_3_NO	[M − H]^−^	198.43	2-Hydroxy-3,5,6-trichloropyridine

**Table 5 toxics-12-00402-t005:** Degradation potential of environmental chlorpyrifos-degrading microorganisms.

Strains	Biodegradation Potential	References
*Hortaea* sp. B15	For chlorpyrifos at 400 mg·L^−1^, 91.1% degradation was achieved in 20 h. The degradation products were 3,5,6-trichloropyridin-2-ol and 2-pyridinol.	[53]
*Bacillus* sp. CP6 and *Klebsiella pneumoniae* sp. CP19	For chlorpyrifos at 250 mg·L^−1^, 93.4 ± 2.8% degradation was achieved in 16 days. Degradation products were not reported.	[47]
*Bacillus* sp. Ct3	For chlorpyrifos at 125 mg·L^−1^, 88% degradation was achieved in 8 days. The degradation product was 3,5,6-trichloro-2-pyridinol (TCP).	[54]
*Sphingobacterium* sp. C1B	For chlorpyrifos at 50 mg·L^−1^, 84% degradation was achieved in 14 days. The degradation products were 3,5,6-trichloro-2-pyridinol to benzene and 1,3-bis (1,1-dimethylethyl).	[55]
*Shewanella* sp. BT05	For chlorpyrifos at 10 mg·L^−1^, the degradation rate was 94.3% at 24 h. The degradation product was 3,5,6-trichloro-2-pyridinol (TCP).	[10]
*Shewanella oneidensis* MR-1	For chlorpyrifos at 19.18 mg·L^−1^, 75.71% degradation was achieved in 10 days. The degradation products were O,O-Diethyl phosphorothionate, Chlorpyrifos methyl oxon, Dechlorination of Chlorpyrifos, Chlorpyrifos-methyl, Chlorpyrifos oxon, O,O-diethyl thiophosphonate, and 2-Hydroxy-3,5,6-trichloropyridine.	This study

## Data Availability

Data are contained within the article and Appendix A.

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
