# Peer review of "Biotransformation of Chlorpyrifos Shewanella oneidensis MR-1 in the Presence of Goethite: Experimental Optimization and Degradation Products"

_toxics, 2024, doi:10.3390/toxics12060402_

Round 1
Reviewer 1 Report
Comments and Suggestions for Authors
The MS submitted by Tang and coworkers is focused on an environmental friendly approach to remove chlorpyrifos (a moderately toxic and moderately persistent organophosphate insecticide) based on the use of a Shewanella oneidensis MR-1 and goethite complex.
The topic of the MS fits the general scope of the journal. It is well-written and organised, and the results explained are well supported by the graphs and tables. However, some concerns must be addressed by the authors to improve this version.
Major issues:
- Materials and Methods sections must be revised to provide all the details needed to make the experiments 100% reproducible by potential readers.
- Some sentences in the introduction must be revised and rewritten according to previous literature on this topic.
- It should be worth comparing the efficiency of the removal of this pollutant by this approach with other previous methods already reported in the literature.
Minor issues:
- Some English grammar and typos might be revised.
- The quality and size of some figures must be improved.
- Guide for authors must be revised for reference format.
Comments have been embedded through the MS to help the authors to improve this MS. I hope you find them useful.
Comments on the Quality of English LanguageMinor issues:
- Some English grammar and typos might be revised.
Author Response
|
Reviewers' Specific comment |
Our response and revision (Revise according to reviewers’ advices) |
|
Materials and Methods sections must be revised to provide all the details needed to make the experiments 100% reproducible by potential readers. |
We have added more detailed experimental steps in line 189-196. |
|
Some sentences in the introduction must be revised and rewritten according to previous literature on this topic. |
We've made some changes in line 62-70. |
|
It should be worth comparing the efficiency of the removal of this pollutant by this approach with other previous methods already reported in the literature. |
Some similar experiments are listed in Table 5. |
|
Some English grammar and typos might be revised. |
We have carefully reviewed the full text and made some changes. Line 49 “through consumption of contaminated drinking water and food” change to “through the consumption of contaminated drinking water and food” Line 104 “However, chlorpyrifos degradation studies should not only consider its initial concentration and degradation efficiency” change to “However, chlorpyrifos degradation studies should consider not only its initial concentration” Line 107 “involved in microbial metabolism” change to “which are involved in microbial metabolism.” Line 112 “When the concentration of toxic pollutants is low” change to “when the concentration of toxic pollutants is low” Line 117 “microorganisms can efficiently transform these toxic pollutants” change to “can microorganisms efficiently transform these pollutants” Line 121 “In addition, coexistence of TCP with parental organisms leads to toxic synergisms” change to “In addition, the coexistence of TCP with parental organisms leads to toxic synergies” Line 148 “was acquired from Shanghai Ampoule Experimental Technology Co., Ltd.” change to “was acquired from Shanghai Anpel Experimental Technology Co., Ltd.” |
|
The quality and size of some figures must be improved. |
We made changes to Figure 2,Figure 3,Figure 5 and Figure 6. |
|
Guide for authors must be revised for reference format. |
Done. |

Reviewer 2 Report
Comments and Suggestions for Authors<Biodegradation of chlorpyrifos by goethite / Shewanella onei- 2 densis MR-1 complex: experimental optimization and degradation products is an important manuscript dedicated to agrochemical degradation. THe paper is well presented; however, some details could be checked.
this organophosphate insecticide, is extensively utilized worldwide, and we have to study its presence in drinking water and degradation. chlorpyrifos can human consumption is trhough drinking water and food
--please, check in Figure 7 legend. Chlorpyrifos degradation productions. / or products?
Please, explain and highlight any control or controls samples.
Comments on the Quality of English LanguageEnglish reading is fine.
Author Response
|
Reviewers' Specific comment |
Our response and revision (Revise according to reviewers’ advices) |
|
Please, check in Figure 7 legend. Chlorpyrifos degradation productions. / or products? |
In Figure 7, compound 6 is chlorpyrifos itself, and the others are degradation products of chlorpyrifos. |
|
Please, explain and highlight any control or controls samples. |
Current studies believe that the main metabolite of chlorpyrifos is 3,5, 6-trichloro-2-Pyridinol (TCP), and the authors agree with this view. Therefore, we did not do a separate controls sample, but focused on exploring the different metabolites of chlorpyrifos under the influence of different external environments. |
|
Comments on the Quality of English Language. |
We have carefully reviewed the full text and made some changes. Line 49 “through consumption of contaminated drinking water and food” change to “through the consumption of contaminated drinking water and food” Line 104 “However, chlorpyrifos degradation studies should not only consider its initial concentration and degradation efficiency” change to “However, chlorpyrifos degradation studies should consider not only its initial concentration” Line 107 “involved in microbial metabolism” change to “which are involved in microbial metabolism.” Line 112 “When the concentration of toxic pollutants is low” change to “when the concentration of toxic pollutants is low” Line 117 “microorganisms can efficiently transform these toxic pollutants” change to “can microorganisms efficiently transform these pollutants” Line 121 “In addition, coexistence of TCP with parental organisms leads to toxic synergisms” change to “In addition, the coexistence of TCP with parental organisms leads to toxic synergies” Line 148 “was acquired from Shanghai Ampoule Experimental Technology Co., Ltd.” change to “was acquired from Shanghai Anpel Experimental Technology Co., Ltd.” |

Round 2
Reviewer 1 Report
Comments and Suggestions for Authors
Thanks for your time addressing all the comments and suggestions that I made
Comments on the Quality of English Languagesome typos